# Clinical Efficacy of Selumetinib in Alleviating Neuropathic Pain Associated with Plexiform Neurofibroma: A Case Series

**DOI:** 10.3390/genes16060645

**Published:** 2025-05-28

**Authors:** Giuseppina Miele, Irene Russo, Lorenzo Filipponi, Silverio Perrotta, Elisabetta Maida, Giulio Piluso, Mariarosa Anna Beatrice Melone, Claudia Santoro

**Affiliations:** 1Department of Advanced Medical and Surgical Sciences, InterUniversity Center for Research in Neurosciences and Center for Rare Diseases, 2nd Division of Neurology, University of Campania “Luigi Vanvitelli”, 80131 Naples, Italy; giuseppina.miele@unicampania.it (G.M.); elisabetta.maida@studenti.unicampania.it (E.M.); 2Department of Women’s and Children’s Health and General and Specialist Surgery, University of Campania “Luigi Vanvitelli”, Via Luigi De Crecchio 2, 80138 Naples, Italy; irenerusso122@gmail.com (I.R.); lorenzo.filipponi@studenti.unicampania.it (L.F.); silverio.perrotta@unicampania.it (S.P.); claudia.santoro@unicampania.it (C.S.); 3Pediatric Neurology, Santobono-Pausilipon Childrens Hospital, 80129 Naples, Italy; 4Department of Precision Medicine, University of Campania “Luigi Vanvitelli”, 80138 Naples, Italy; giulio.piluso@unicampania.it; 5Sbarro Institute for Cancer Research and Molecular Medicine, Center for Biotechnology, Temple University, Philadelphia, PA 19122, USA

**Keywords:** neuropathic pain, selumetinib, MEK inhibitor, neurofibromatosis type 1, plexiform neurofibroma

## Abstract

**Background**: Selumetinib, an MEK inhibitor, was approved by the U.S. Food and Drug Administration in April 2020 and became reimbursable in Italy in January 2020, for treating patients aged ≥3 years with neurofibromatosis type 1 (NF1) complicated by symptomatic, inoperable plexiform neurofibromas (PNs). Selumetinib has been shown to effectively reduce the volume of target PNs and alleviate neuropathic pain even in long-term-treated patients. We report the impact of Selumetinib on pain in three NF1 cases with inoperable symptomatic plexiform neurofibromas. **Case Series**: Three patients with NF1 (aged 13–27 years) presented with symptomatic, inoperable plexiform neurofibromas (PNs) associated with severe neuropathic pain. Following selumetinib administration, a marked reduction in or disappearance of neuropathic pain was observed within a few weeks, allowing the complete discontinuation of pain therapy. Interestingly, pain recurred whenever selumetinib was temporarily suspended, requiring patients to resume analgesic therapy. **Conclusions**: Selumetinib treatment has been shown to effectively reduce neuropathic pain in patients with NF1. These findings represent a significant advancement in managing pain related to PNs and support its potential application in treating other forms of neuropathic pain.

## 1. Introduction

Neurofibromatosis type 1 (NF1) is an autosomal dominant genetic disorder caused by mutations in the *NF1 gene (17q11.2)*, leading to a wide spectrum of systemic manifestations, with predominant, though not exclusive, involvement of the nervous system. Plexiform neurofibromas (PNs) are a common manifestation in NF1 patients. These benign nerve sheath tumors affect up to 50% of patients and have the potential for rapid growth and malignant transformation in 8–15.8% of cases [1]. The consequences can be severe, including disfigurement, visual impairment, loss of mobility, bladder dysfunction, hemorrhage, spinal cord compression, and neuropathic pain, which occurs in up to 60% of NF1 patients with PNs [1,2]. Moreover, PNs can infiltrate nerves, blood vessels, and organs, making surgical resection unfeasible due to the risk of significant morbidity [2,3].

Selumetinib, an MEK inhibitor, was approved by the FDA in April 2020 and in Italy by AIFA in January 2024 in patients aged ≥3 years with NF1 complicated by symptomatic, inoperable PNs [4,5]. The recommended dose of selumetinib is 25 mg/m^2^ of body surface area (BSA), administered orally twice daily (BID) in continuous 28-day treatment cycles. Consequently, managing PN-associated neuropathic pain remains challenging, as effective and sustained pain control is frequently difficult to achieve. Furthermore, neuropathic pain in NF1 can also occur independently of PNs and may result from a form of peripheral neuropathy associated with NF1, characterized by widespread axonal atrophy and degeneration in the absence of tumor infiltration. This may be due to an impairment in both Schwann cells (SCs) and SC–fibroblast cross-talk [6].

This paper provides insights into the use of selumetinib for severe neuropathic pain due to PNs in patients with NF1. We present three cases of NF1 patients with symptomatic, inoperable PNs and severe neuropathic pain who experienced a marked reduction in or complete resolution of symptoms within weeks of starting selumetinib, leading to the suspension of concomitant analgesic therapies. All of them received the drug on a compassionate-use basis prior to 2020.

## 2. Case Series

### 2.1. Case 1

An 18-year-old male with sporadic NF1 was diagnosed at the age of 11. His clinical course was complicated by the development of a giant PN, which was inoperable due to its location. The tumor originated from the lumbosacral plexus and extended along the entire left lower leg, leading to a significant overgrowth of the ipsilateral iliac wing and lower leg, as documented radiologically and clinically.

From January 2017 to January 2019, the patient underwent a series of surgical procedures to address complications associated with the PN. These interventions included PN debulking, femoral epiphysiodesis, and removal of the tibial and femoral plates. Despite these measures, the patient continued to experience persistent neuropathic pain, necessitating the initiation of pain management therapy in August 2019. The therapy consisted of oxycodone 0.5 mg daily and gabapentin 100 mg BID.

However, his condition rapidly deteriorated, severely limiting his ability to sit and walk [Eastern Cooperative Oncology Group (ECOG) performance status = grade 3; Riccardi scale score = 4] [7,8]. Additionally, his sleep was severely disrupted, which ultimately resulted in school discontinuation. The volume of the target PN was measured at 2794 cm^3^. At the age of 13, in August 2019, he began treatment with selumetinib due to the limited success of surgical interventions and poor pain control [Visual Analog Scale (VAS) = 8]. The patient experienced substantial pain relief, leading to complete discontinuation of analgesic therapy after two cycles.

Shortly after starting selumetinib treatment, he developed bilateral great toenail paronychia (grade 1), which was treated with a course of topical 0.05% clobetasol BID and topical mupirocin BID for two weeks. The paronychia rapidly progressed to grade 2 on the Common Terminology Criteria for Adverse Events (CTCAE) [9], affecting other toes and fingers. Local therapy was maintained, and oral antibiotics (amoxicillin + clavulanic acid 1 g BID) and prednisone (initially 5 mg, subsequently reduced to 2.5 mg daily) were introduced. Ultimately, selumetinib was discontinued after 16 weeks due to the persistence of paronychia and the emergence of a rash on the trunk and lower extremities. Analgesic therapy was resumed the following week with 1000 mg paracetamol BID.

Due to the restrictions imposed by the global pandemic caused by the SARS-CoV-2 virus, selumetinib therapy was restarted 53 weeks after its discontinuation. At that time, the patient exhibited an ECOG performance status of grade 3 despite ongoing analgesic therapy. Upon restarting selumetinib, there was a marked improvement in pain, and by the fifth week of therapy, all analgesic treatments were permanently discontinued. The patient continued to receive selumetinib monotherapy, with radiological confirmation of a 41% reduction in target PN volume documented at the 38th cycle, in comparison to the second restart date. The patient exhibited complete pain control (VAS = 0) from a clinical perspective.

In March 2024, he underwent an orthopedic surgical procedure unrelated to the PN, for which selumetinib was discontinued three days prior. One week after the cessation of the pharmacological intervention, the patient experienced a recurrence of neuropathic pain necessitating the administration of a combination of oxycodone 10 mg BID, gabapentin 300 mg BID, and paracetamol 1000 mg three times daily (TID). This regimen provided partial pain relief. Selumetinib was withheld for one month, during which the patient continued the analgesic therapy. Upon resuming selumetinib at full dosage, only paracetamol 1000 mg BID was required during the first week, after which selumetinib was continued as monotherapy, with no further pain reported. The patient is currently undergoing continuous selumetinib therapy, with no reported pain, an ECOG performance status of grade 2, and a Riccardi scale score of 2.

### 2.2. Case 2

A 28-year-old male, diagnosed with NF1 at the age of 6, developed a thalamic glioma that was treated with chemotherapy and radiation therapy, leading to post-actinic angiitis. He presented with a PN in the right lower limb (ECOG performance status = grade 4, Riccardi scale score = 3/4), which involved the right intervertebral foramen between S1 and S2 and subsequently extended into the pelvic cavity, following the anatomical course of the sciatic nerve to the posterior compartment of the right thigh.

The PN, monitored radiologically and clinically, showed progressive enlargement, and neuropathic pain developed. The pain progressively worsened, prompting the initiation of pain management therapy at the age of 22. The patient was prescribed pregabalin 75 mg BID and paracetamol 500 mg with codeine 30 mg as needed, which was later increased to paracetamol 500 mg plus codeine 30 mg TID (VAS = 9/10).

In September 2022, at the age of 27, he started treatment with selumetinib due to the increased size of the right limb PN and escalating pain, with the tumor being inoperable. After initiating MEK inhibitor therapy, the pain showed rapid and substantial regression (VAS = 0/10), leading to the discontinuation of all analgesic medications within a few weeks. The patient developed an acneiform rash, which did not necessitate dose adjustment or treatment interruption.

In July 2023, due to a delay in drug resupply, the patient experienced a recurrence of pain (VAS = 7/10), and the patient had to resume paracetamol 500 mg plus codeine 30 mg TID for two weeks. This regimen was maintained until two days after selumetinib was reintroduced. Although the volume of the PN remained unchanged, cerebral MRI revealed a significant reduction in edema surrounding the right thalamic neoplasm, with the maximum lesion diameter reduced from approximately 5 cm to 4.3 cm.

### 2.3. Case 3

An 18-year-old female with maternally inherited NF1, diagnosed at the age of 3, presented with an inoperable paravertebral PN originating from the left lumbosacral plexus and extending along the entire left leg, associated with neuropathic pain (VAS = 7, ECOG performance status = grade 1, Riccardi scale score = 3/4). The pain was managed with paracetamol 1000 mg as needed.

At 15 years of age, she underwent total excision of an intrinsic nodularity in the PN demonstrating increased metabolic activity on PET imaging (pathological SUV), which histologically confirmed an atypical neurofibroma. In December 2021, she reported a worsening lumbosacral pain (VAS = 8, ECOG performance status = grade 3) and was treated with gabapentin 300 mg TID and paracetamol 1000 mg TID. Three months later, she started therapy with selumetinib. Following the initiation of selumetinib, analgesic therapy was withdrawn within ten days of treatment initiation (VAS = 0, ECOG performance status = grade 1).

Selumetinib therapy resulted in stable disease (documented at cycle 22) and complete neuropathic pain control until cycle 42, when radiological progression (>20% increase in volume compared to baseline) of the target lesion, as well as of a contralateral PN, was noted and the treatment was suspended. A few days later, she began complaining of worsening neuropathic pain (VAS = 7/10, ECOG = grade 2/3) and functional impairment, requiring antalgic therapy. Four months after discontinuing selumetinib, a good control of pain (VAS = 4/10) was achieved with gabapentin, 900 mg daily, duloxetine 60 mg daily, and a neuroactive phytotherapeutic compound.

A summary of the three cases and their main clinical features is provided in Table 1.

## 3. Discussion

Our study confirms that selumetinib is an effective treatment for patients with NF1 with symptomatic and inoperable PNs. Selumetinib was provided for compassionate use and on a named-patient basis by AstraZeneca and approved by the local ethics committee in all reported cases. All three patients experienced a marked reduction in or complete resolution of neuropathic pain within weeks of initiating therapy.

The pain associated with NF1 appears to have two primary causes: the first is directly related to the presence of PNs and their compression of surrounding nerves, and the second is linked to neurofibromin dysfunction. As patients with NF1 commonly experience increased pain sensitivity, several studies have explored the role of neurofibromin in modulating sensory neuron excitability. Basic science research has indicated that neurofibromin deficiency affects nociceptive sensory neurons. For instance, heterozygous NF1 mutant mice show increased sensitivity to chemical stimuli, as demonstrated by elevated neuropeptide release following stimulation [10]. Additionally, small-diameter sensory neurons from NF1+/− mice exhibit increased expression of N-type calcium channels compared to their wild-type counterparts [11], further contributing to aberrant sensory neuron signaling. It has been hypothesized that abnormal intercellular signaling between these cells could be central to the development of NF1-related neuropathies [12], although direct experimental evidence is still lacking.

However, several animal models have been proposed for the purpose of investigating NF1-associated pain. In one such study, Moutal et al. identified a novel interaction between the cytosolic protein collapsin response mediator protein 2 (CRMP2) and neurofibromin [13]. CRMP2, which regulates the activity of ion channels such as CaV2.2 and NaV1.7, has been shown to interact directly with CaV2.2, thereby enhancing calcium influx and neurotransmitter release in sensory neurons.

As postulated by [13], the wild-type neurofibromin has been demonstrated to inhibit CRMP2, consequently reducing calcium influx and subsequent pain transmission. However, the truncated form of neurofibromin is incapable of performing this function, resulting in heightened pain sensitivity. Furthermore, CRMP2 has been shown to interact with the C-terminal domain of neurofibromin. The loss of this interaction results in increased CRMP2 phosphorylation, which in turn promotes its association with ion channels. Furthermore, CRMP2 has been demonstrated to regulate NaV1.7, a pivotal factor in pain perception, whose activity has been observed to increase in NF1 [13]. These findings highlight the intricate relationship among calcitonin gene-related peptide (CGRP), CRMP2, neurofibromin, and ion channels in NF1-related pain pathways (Figure 1a).

To date, no clinical or preclinical studies have been published that directly investigate the interaction between selumetinib, a selective MEK1/2 inhibitor, and CGRP in models of pain or migraine. Nonetheless, circumstantial evidence suggests a possible interaction between the MEK/ERK signaling pathway and CGRP-mediated mechanisms, particularly in the context of nociceptive processing.

Indeed, preliminary clinical studies suggest that the MEK/ERK signaling pathway may influence CGRP expression and contribute to pain processing, indicating a potential link between these two systems. In a rat model of bladder inflammation, nerve growth factor (NGF) triggered an increase in calcitonin gene-related peptide (CGRP) levels in sensory neurons through the activation of the extracellular signal-regulated kinase 5 (ERK5) pathway [14]. This effect was blocked by MEK pathway inhibitors, thereby demonstrating that MEK/ERK signaling can regulate CGRP production [14] (Figure 1b). In addition, other studies have demonstrated that the suppression of MEK/ERK activity leads to a reduction in both CGRP levels and pain sensitivity in models of inflammatory visceral pain [15]. 

Although this interaction has not been specifically studied in NF1 patients treated with selumetinib, these findings suggest that MEK inhibition may also affect CGRP-related mechanisms in this context. This could have implications for pain modulation in NF1 patients. Further findings demonstrate the critical importance of the MEK/ERK signaling pathway in regulating neuronal excitability and pain perception. In the context of mouse models of neuropathic pain, MEK inhibition has emerged as a promising therapeutic approach [16,17,18]. 

Activation of this pathway by pituitary adenylate cyclase-activating polypeptide (PACAP) has been shown to enhance neuronal excitability, in part by phosphorylating voltage-dependent Na+ channels such as NaV1.7. This process is particularly evident in conditions such as chronic constriction injury of the infraorbital nerve (ION-CCI) in rats, where PACAP-mediated activation of MEK/ERK leads to increased pain sensitivity, known as allodynia. In patients with NF1, abnormal activation of the MEK/ERK pathway not only contributes to tumorigenesis but may also exacerbate pain symptoms through similar mechanisms [19].

Available evidence suggests that MEK inhibition may induce a depolarizing shift in NaV1.7 activation, thereby reducing neuronal excitability and potentially alleviating pain.

Furthermore, it should be noted that the manifestation of neuropathic pain in patients diagnosed with NF1 can occur independently of the presence of PNs. This may be attributable to a form of peripheral neuropathy that is concomitant with NF1, which is typified by widespread axonal atrophy and degeneration in circumstances where there has been no infiltration of tumor cells. The underlying mechanism may be attributed to an impairment in both Schwann cells (SCs) and SC–fibroblast cross-talk [6]. From a neuropathological perspective, the concept of neurofibromas as ‘unrepaired wounds’ has gained ground following studies in a mouse model. These studies found that peripheral nerve injury and NF1 gene deficiency in Schwann cells promote neurofibroma growth and nerve thickening [20]. Nevertheless, the link between these structural changes and the pathogenesis of NF1-related neuropathies remains unclear [21].

Our findings demonstrate that the early reduction in neuropathic pain observed in our three cases, even prior to the significant volumetric reduction of the PNs, can be attributed to the direct biological effect of selumetinib on the pain pathways. In these cases, neuropathic pain therapy was discontinued within a very short time (less than one month) after treatment initiation, well before any substantial reduction in PN volume.

This supports the hypothesis that biochemical and functional changes may precede macroscopic modifications, exerting a positive impact on the pain pathway from the outset. It also suggests that neurofibromatosis-related pain may be at least partially independent of the volume and mass effect of neurofibromas themselves.

Several studies have documented the impact of selumetinib on neuropathic pain [22] in addition to its ability to induce objective regression or stabilization of PNs [19]. A Phase I trial of selumetinib in children with NF1 and inoperable plexiform neurofibromas demonstrated unequivocal efficacy, with 8 out of 13 patients reporting improvement in PN-related neuropathic pain [3]. A single-center study conducted by Espírito Santo et al. revealed that all four patients affected by PN-related neuropathic pain exhibited a positive response to the drug [23]. Moreover, a recent meta-analysis, including 10 clinical trials with 268 patients, demonstrated selumetinib’s effectiveness in treating NF1-associated plexiform neurofibromas (PNs). Of these, six studies involving 67 patients reported a pain improvement rate of 75.3%, using a random-effects model, highlighting its positive impact on pain management [24].

Regarding the safety profile of selumetinib, the side effects experienced by our patients were common and manageable, and they were treated in line with the current consensus [25].

The safety profile of the drug has been explored in long-term-treated patients belonging to Phase 1 and Phase 2 stratum 1 of the SPRINT study [22]. The authors observed a sustained radiological response over a five-year period of additional selumetinib treatment, with a continued improvement in pain beyond what had previously been reported at one year. In a similar vein, the therapeutic regimen administered to patients comprised up to 42 cycles without any recorded issues pertaining to an escalation in adverse event frequency or severity, as evidenced by case 3. Based on our experience and the previously reported pain-relieving effects of selumetinib, we are convinced that MEK inhibitors represent a promising therapeutic approach capable of not only reducing or stabilizing PN volume but also effectively controlling chronic neuropathic pain, thereby significantly enhancing patients’ quality of life (QoL).

Nevertheless, further research is required, particularly through larger, randomized controlled trials, to validate these findings and to fully explore the broader implications of MEK inhibition in the treatment of PN-related symptoms. Future research should also prioritize comparative studies to evaluate the effectiveness of selumetinib against other current treatment options, including both pharmacological treatments (such as pain-relieving medications) and surgical interventions. Such comparative analysis will provide valuable insights into the efficacy, side effects, and overall impact on QoL, aiding clinicians in developing more tailored, efficacious treatment plans for patients with NF1.

## Figures and Tables

**Figure 1 genes-16-00645-f001:**
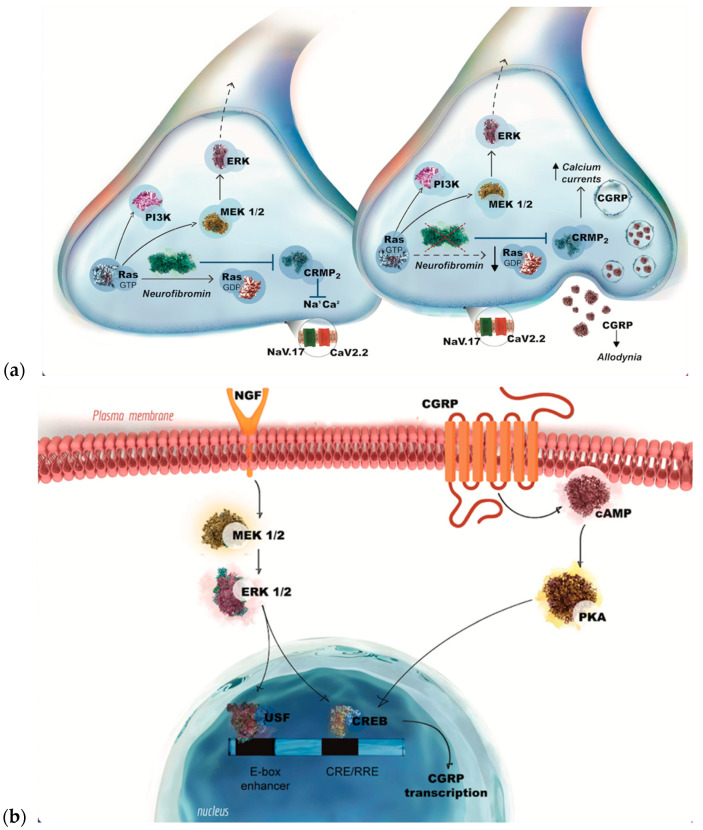
(**a**) The role of neurofibromin in modulating the MEK/ERK and CGRP pathways in NF1. (**b**) MEK/ERK pathway as a potential regulator of CGRP expression in NF1-associated pain.

**Table 1 genes-16-00645-t001:** Clinical and therapeutic characteristics of three NF1 patients with symptomatic, inoperable plexiform neurofibromas treated with selumetinib.

		Case 1	Case 2	Case 3
	Sex/Age When Starting Selumetinib	Male, 18 y	Male, 28 y	Female, 18 y
	NF1 Diagnosis Age	11 years	6 years	3 years
	NF1 (NM_000267.3) Pathogenic Variant	c.2681T>C(p.Phe894Ser)	c.1198C>T(p.Gln400Ter)	c.1198C>T(p.Arg416Ter)
	PN Location	Lumbosacral plexus + left lower limb	S1–S2 nerve roots → right thigh	Left lumbar plexus + left lower limb
Time 0	ECOG Performance Status	3	4	1 → 3
	Worse VAS value	8	9	7 → 8
	Analgesics Drugs	Oxycodone + Gabapentin	Pregabalin + Paracetamol/Codeine	Gabapentin + Paracetamol
During Selumetinib Treatment	ECOG Performance Status	2	4	1
	Worse VAS value	0	0	0
	Analgesic drugs	None after 2 cycles	None after few weeks	None after 10 days
	Main Side Effects	Paronychia → temporary interruption	Acneiform rash (managed)	None significant
	Radiological Response	−41% PN volume at cycle 38	Stable PN, ↓ thalamic edema	Stable PN until cycle 42, then >20% volume increase → treatment discontinued
After Selumetinib Discontinuation	ECOG Performance Status	Treatment still ongoing	Treatment still ongoing	2/3
	Worse VAS value	Treatment still ongoing	Treatment still ongoing	4
	Analgesics Drugs	Treatment still ongoing	Treatment still ongoing	Gabapentin + duloxetine + nutraceutic

NF1, neurofibromatosis type 1; PN, plexiform neurofibroma; VAS, Visual Analog Scale; ECOG, Eastern Cooperative Oncology Group; BID, twice daily; TID, three times daily; CTCAE, Common Terminology Criteria for Adverse Events.

## Data Availability

The data presented in this study are available on request from the corresponding author due to clinical nature and the need to protect patient confidentiality.

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
