# Peer review of "Clinical Efficacy of Selumetinib in Alleviating Neuropathic Pain Associated with Plexiform Neurofibroma: A Case Series"

_genes, 2025, doi:10.3390/genes16060645_

Round 1
Reviewer 1 Report
Comments and Suggestions for Authors
The authors present three cases where severe plexiform neurofibromas were treated with MEK inhibitor. The authors postulate that the initiation of the MEK inhibitor ameliorated pain, and propose some mechanistic pathways in the discussion. The article is a well-written accumulation of case reports and adds to the body of literature on MEK inhibition. A few suggestions for improvement include:
- The NF1 gene mutation of the three cases should be added to the table.
- Aside from patient self-reported questionnaires, were there any objective measurements of neuropathy such as nerve conduction studies
- Was there any documentation of the response being independent of the plexiform size responses (e.g. symptom resolution prior to radiologic response).
Author Response
Dear Editor,
Thank you for giving us the opportunity to submit a revised draft of our manuscript titled "Clinical Efficacy of Selumetinib in Alleviating Neuropathic Pain Associated with Plexiform Neurofibroma: A Case Series" to Genes_Section:Human Genomics and Genetic Diseases_Special issue:Rare Neurogenetic Disorders in the Third Millennium: Diagnostic and Therapeutic Challenges.
We would like to express our gratitude for the time and effort you and the reviewers have invested in providing your valuable feedback on our manuscript.
In addition, we gratefully acknowledge the insightful comments provided by the reviewers in their assessment of our paper. We have incorporated changes in line with the suggestions made by the reviewers.
We have highlighted the changes within the manuscript.
Please find below a point-by-point response to the reviewers' comments and concerns.
Comments from Reviewer 1
Comments:The authors present three cases where severe plexiform neurofibromas were treated with MEK inhibitor. The authors postulate that the initiation of the MEK inhibitor ameliorated pain, and propose some mechanistic pathways in the discussion. The article is a well-written accumulation of case reports and adds to the body of literature on MEK inhibition. A few suggestions for improvement include:
- The NF1 gene mutation of the three cases should be added to the table.
As per your suggestion, we have added the data to the table. Professor Giulio Peluso is responsible for the molecular data. In instances where laboratory data is included in our articles, the name of the laboratory member responsible for the analysis and research on our patients is always reported. I am pleased to inform you that, following agreement from all authors, Professor Piluso and his affiliation have been included in the revised version of the manuscript. Professor Piluso has reviewed the revised manuscript and provided his approval.
- Aside from patient self-reported questionnaires, were there any objective measurements of neuropathy such as nerve conduction studies.
We are in agreement with the reviewer that electrophysiological studies may contribute to the clinical characterisation of polyneuropathy in NF1 patients. In Neurofibromatosis 1 (NF1), neuropathic pain can correlate with both polyneuropathies and plexiform neurofibromas. While plexiform neurofibromas can cause pain through pressure on nerves and organs, the nerve damage associated with NF1, including polyneuropathies, can also lead to neuropathic pain. An ultrastructural study conducted by us in 2018 documented widespread axonal atrophy and degeneration in neurofibromatous neuropathy, occurring independently of the presence of tumour infiltration, an impairment in SC (Schwann cells)-axon cross talk (doi.org/10.1080/01913123.2018.1454562). This aspect has been thoroughly discussed in the manuscript.
However, in focal neuropathies, whether traumatic or due to nerve entrapment, MRI imaging has improved the diagnostic accuracy by directly visualising underlying nerve lesions and providing information on the exact lesion localization, extension, and spatial distribution, thereby assisting surgical planning.
In the case of our patients, nerve conduction studies documented mild sensory motor polyneuropathy in the initial clinical phase. An imaging diagnosis (MRI) provided more accurate data on the location and characteristics of the lesions.
In subsequent follow-ups, the primary objective was to evaluate the patient's pain symptoms. Following careful consideration, we determined that conducting additional electrophysiological studies was not required. Instead, we performed meticulous follow-up MRI scans to ensure comprehensive monitoring.
Was there any documentation of the response being independent of the plexiform size responses (e.g. symptom resolution prior to radiologic response).
We confidently reported on how pain was measured using dedicated scales shortly after therapy began. This means that any volume effect could have been present. It is well established that the median time for a volumetric response in PN treated with selumetinib is 12 cycles. It is clear that pain reappeared after even very short periods of suspension, which indicates that this was due to a completely independent effect unrelated to any change in tumour volume.
Best Regards
Prof. Dr. Mariarosa AB Melone
InterUniversity Center for Research in Neurosciences
University of Campania Luigi Vanvitelli
Neurology Unit and Center for Rare Diseases
Via Sergio Pansini, 5, Building n. 10
80131 Naples, Italy
& Sbarro Institute Temple University Philadelphia USA
email: marina.melone@unicampania.it

Reviewer 2 Report
Comments and Suggestions for Authors
Selumetinib is an oral, targeted treatment option known to shrink PN and FDA approved for treatment in patients > 2 years of age with NF1 who have plexiform neurofibromas that cannot be surgically excised completely.
Authors have mentioned chronic paronychia resistant to treatment in the first patient. Most patients with neuropathic pain are likely to require the medication over a long period of time, may be their life-time. Suggest authors discuss the side-effects and potential risks associated with the long-term use of this medication.
• What specific improvements should the authors consider regarding the methodology? - authors should discuss the side-effects and potential risks associated with the long-term use of this medication
- Title mismatch : Mismatch between title submitted in the manuscript versus in review report form
Author Response
Dear Editor,
Thank you for giving us the opportunity to submit a revised draft of our manuscript titled "Clinical Efficacy of Selumetinib in Alleviating Neuropathic Pain Associated with Plexiform Neurofibroma: A Case Series" to Genes_Section:Human Genomics and Genetic Diseases_Special issue:Rare Neurogenetic Disorders in the Third Millennium: Diagnostic and Therapeutic Challenges.
We would like to express our gratitude for the time and effort you and the reviewers have invested in providing your valuable feedback on our manuscript.
In addition, we gratefully acknowledge the insightful comments provided by the reviewers in their assessment of our paper. We have incorporated changes in line with the suggestions made by the reviewers.
We have highlighted the changes within the manuscript.
Please find below a point-by-point response to the reviewers' comments and concerns.
Comments from Reviewer 2
Comments: Selumetinib is an oral, targeted treatment option known to shrink PN and FDA approved for treatment in patients > 2 years of age with NF1 who have plexiform neurofibromas that cannot be surgically excised completely.
Authors have mentioned chronic paronychia resistant to treatment in the first patient. Most patients with neuropathic pain are likely to require the medication over a long period of time, may be their life-time. Suggest authors discuss the side-effects and potential risks associated with the long-term use of this medication.
- What specific improvements should the authors consider regarding the methodology? - authors should discuss the side-effects and potential risks associated with the long-term use of this medication
We would like to express our gratitude for the commentary provided. In response, we have incorporated a concise paragraph in the discussion section that addresses the subject matter, accompanied by the pertinent references.
- Title mismatch : Mismatch between title submitted in the manuscript versus in review report form
Rest assured, we have corrected the mismatch.
Best Regards
Prof. Dr. Mariarosa AB Melone
InterUniversity Center for Research in Neurosciences
University of Campania Luigi Vanvitelli
Neurology Unit and Center for Rare Diseases
Via Sergio Pansini, 5, Building n. 10
80131 Naples, Italy
& Sbarro Institute Temple University Philadelphia USA
email: marina.melone@unicampania.it
